# Wnt Signaling and Aging of the Gastrointestinal Tract

**DOI:** 10.3390/ijms232012210

**Published:** 2022-10-13

**Authors:** Naoki Asano, Akio Takeuchi, Akira Imatani, Masashi Saito, Xiaoyi Jin, Waku Hatta, Kaname Uno, Tomoyuki Koike, Atsushi Masamune

**Affiliations:** Division of Gastroenterology, Tohoku University Graduate School of Medicine, Sendai 980-8575, Japan

**Keywords:** aging, stem cell, stem cell aging, Wnt, beta-catenin, stomach

## Abstract

Aging is considered a risk factor for various diseases including cancers. In this aging society, there is an urgent need to clarify the molecular mechanisms involved in aging. Wnt signaling has been shown to play a crucial role in the maintenance and differentiation of tissue stem cells, and intensive studies have elucidated its pivotal role in the aging of neural and muscle stem cells. However, until recently, such studies on the gastrointestinal tract have been limited. In this review, we discuss recent advances in the study of the role of Wnt signaling in the aging of the gastrointestinal tract and aging-related carcinogenesis.

## 1. Introduction

Aging is one of the most intriguing themes of research in the field of life science, especially in this aging society, since it causes deterioration of organic functions and leads to various diseases, including life-threatening cancers [1]. Understanding the molecular mechanisms underlying aging will help us to live better and healthier lives. Various mice models have been employed to study aging in vivo [2]. The Klotho-deficient mouse is an aging mice model that mimics progeria, exhibiting accelerated aging [3]. Studies of Klotho-deficient mice have demonstrated that Klotho performs as an antagonist of Wnt signaling and that continuous Wnt stimulation, as a result of Klotho deficiency, can accelerate aging [4,5]. The finding that ceaseless Wnt stimulation leads to accelerated aging has been supported by other reports that have evidenced persistent Wnt stimulation leading to exhaustion of hematopoietic and muscular stem cells [6,7,8].

Wnt signaling is a major signaling pathway that has been intensively studied [9,10,11]. There are two main pathways activated by secreted Wnt ligands, namely, the canonical and non-canonical pathways. Of the two pathways, the canonical Wnt signaling pathway involves β-catenin activation, whereas the non-canonical pathway is activated independent of β-catenin. Since the canonical pathway is highly conserved in evolution and has been shown to be important in the molecular mechanism of aging in various organs, as described later, we will focus on this pathway.

In the absence of the Wnt ligand, cytosolic β-catenin binds to the β-catenin destruction complex, which includes the scaffolding protein Axin, adenomatous polyposis coli (APC), serine/threonine kinase glycogen synthesis kinase 3 (GSK3), and casein kinase 1 (CK1). The β-catenin is then phosphorylated and becomes ubiquitinated by E3-ubiquitin ligase β-transducin repeat containing protein (β-TrCP), and, finally, is subjected to proteasomal degradation [12]. On the other hand, binding of Wnt ligands to their receptor, which consists of the Frizzled and low-density lipoprotein receptor related protein 5/6 (LRP5/6), activates Dishevelled (Dvl) and induces dissociation of the destruction complex, allowing the cytosolic β-catenin to escape from proteasomal degradation. β-catenin will then translocate into the nucleus, bind to the T-cell factor (or transcription factor)/lymphoid enhancer factor (TCF/LEF), and activate its downstream genes (Figure 1) [10,13]. Inhibitors of Wnt signaling, such as Dickkopf (DKK), secreted Frizzled-related protein (sFRP), and Notum, take part in regulating aging, as discussed later.

In the gastrointestinal tract, Wnt and other Wnt-related factors are considered to be secreted mainly from mesenchymal cells located in proximity to gastrointestinal stem cells [14]. In response to the secreted Wnt or Wnt-related factors, the Wnt/β-catenin signaling pathway in the gastrointestinal stem cells become activated or suppressed, leading to downstream events.

The role of this canonical Wnt signaling in the development of cancer has been intensively studied [15,16]. Previous studies have reported its role in regulating aging in neural and muscular tissues [17,18,19]. However, until recently, there has only been limited study of its role in the aging of the gastrointestinal tract. Therefore, in this review, we discuss recent advances in the role of Wnt signaling in the aging of the gastrointestinal tract.

## 2. Wnt Signaling and Stem Cell Aging

One of the main causes of aging is the aging of tissue stem cells [20]. In the normal state, stem cells proliferate and differentiate to generate tissue. However, with increasing age, dysfunction of stem cells, such as decreased proliferation and/or skewing in differentiation, starts to occur, which contributes to the initiation of aging-related diseases. This age-related deterioration of stem cell function is regarded as stem-cell aging and is now considered an essential cause of aging. Intensive studies of stem-cell aging have been performed on neural tissue and skeletal muscle. These studies suggest that the Wnt signaling pathway plays a pivotal role in stem-cell aging.

Cho and colleagues have shown that inhibition of sFRP3, a Wnt antagonist previously reported to accumulate and suppress quiescent neural stem cells in the adult brain [21], counteracts neural stem cell aging-related neural disorders observed in genetically engineered mice which exhibit accelerated aging [22]. Similarly, Kalamakis and colleagues reported that sFRP5, another secreted Wnt antagonist, acts together with niche-derived inflammatory signals to induce quiescence of stem cells in the aged murine brain [18]. Another report from Seib and colleagues showed that the expression of Dkk1, also a Wnt antagonist, increased with age in neural progenitor cells in the murine hippocampus and led to declined cognitive function [23]. They found that neural progenitor cells of genetically engineered mice that lacked Dkk1 in the brain showed increased Wnt activity and enhanced self-renewal. In addition, mice that lacked Dkk1 in the brain did not show the age-related deterioration seen in aged Dkk1-intact mice. Taken together, these studies indicated that stem cell aging of neural progenitor cells was due to decreased Wnt signaling.

With respect to studies of stem cell aging in skeletal muscle, Brack and colleagues reported that muscle stem cells of aged mice tended to convert to fibrogenic lineages as a result of enhanced canonical Wnt signaling and that this tendency could be suppressed by Wnt inhibitors [8]. Similarly, Ahrens and colleagues reported correlation of accelerated aging with enhanced Wnt activation [24]. They showed that muscle stem cell numbers and function were impaired upon loss of Klotho expression. They also showed that stem cell function in aged stem cells could be restored by the supplementation of recombinant Klotho protein. Hence, these studies suggested that stem cell aging in muscle stem cells was due to enhanced Wnt activation.

Hair follicles are another tissue in which stem cells have been vigorously investigated. Castilho and colleagues showed that, although activation of Wnt signaling induced growth of hair follicles, prolonged exposure to the Wnt ligand led to exhaustion of stem cells and hair loss, suggesting that enhanced Wnt signaling induced stem cell aging in hair follicular stem cells [25].

The preceding studies have demonstrated that suppressed or enhanced canonical Wnt signaling regulates stem cell aging in neural, muscular, and hair tissues. Similarly, it has become evident that Wnt signaling plays an important role in gastrointestinal stem-cell aging which we discuss below.

## 3. Wnt Signaling and Gastrointestinal Stem Cells

The studies described have shown that Wnt signaling plays a crucial role in regulating the aging of neural, muscular, and hair follicular stem cells. Next, we consider the role of Wnt signaling in regulating gastrointestinal stem cells.

It is now widely accepted that leucine-rich orphan G-protein coupled receptor 5 (Lgr5) is a marker for intestinal stem cells residing at the base of intestinal crypts [26]. Lgr5 was originally discovered as a Wnt target gene expressed in the base of crypts, but not expressed in the villi, of the murine small intestine. Detailed studies have shown that Lgr5+ columnar base cells in the small intestine actively proliferate and differentiate to all lineages of cells that constitute crypts, which has demonstrated that Lgr5+ cells are the stem cells of the small intestine.

Lgr5 is a G-protein-coupled receptor that serves as a receptor for R-spondins [27]. R-spondins are secreted proteins that are considered amplifying regulators of the canonical Wnt signaling pathway by antagonizing DKK1-mediated Wnt inhibition [28]. It has also been shown that the complex of Lgr5 and R-spondin neutralizes transmembrane E3 ubiquitin ligases, which removes Wnt receptors from the surface of stem cells [29]. These reports suggest that Wnt signaling is enhanced in Lgr5+ stem cells. Apart from Lgr5, there are other candidate molecules that have been reported as intestinal stem cell markers, such as Olfactomedin 4 (Olfm4), Musashi 1 (Msi1), Achaete-scute family BHLH transcription factor 2 (Ascl2), leucine-rich repeats and immunoglobulin-like domains 1 (Lrig1), and SPARC-related molecular calcium binding 2 (Smoc2) [30,31].

Stem cell markers in the stomach have been reported to differ between the antrum and the corpus. In the antrum, Lgr5, Axin2, and Aquaporin5 (AQP5) have been reported as markers for gastric stem cells located in the base of antral glands [32,33,34], whereas Bmi1 has been reported as the antral stem cell marker located in the isthmus [35]. In the corpus, on the other hand, Bmi1 and Mist1 have been reported as markers of gastric stem cells which exist in the isthmus of fundic glands [35,36]. In addition, Lgr5+ chief cells [37] and Tnfrsf19+ chief cells [38] in the base of the glands have been reported to behave as reserve gastric stem cells and provide all lineages of the gastric epithelium in situations such as mucosal injuries. The candidates for gastric stem cell marker listed above are molecules related to Wnt signaling, as was the case with the intestinal stem cell markers.

Hence, although the stem cell marker for gastric stem cells is still controversial [39], these earlier reports suggest that gastric stem cells are regulated by Wnt signaling. In addition, recent advances in the use of an organoid culturing system, a technique to expand stem cells in vitro using a 3D-gel culturing method, have shown that both intestinal and gastric stem cells can be expanded in vitro by supplementing Wnt and R-spondins [40], which is further evidence that Wnt signaling plays a pivotal role in the maintenance and differentiation of these cells. Since aging is initiated by the age-related deterioration of stem cells, it follows that Wnt signaling plays an essential role in the aging of the gastrointestinal tract.

## 4. Wnt Signaling and Aging of the Gastrointestinal Tract

So, how does Wnt signaling affect aging of the gastrointestinal tract? A study performed by Nalapareddy and colleagues investigated the influence of aging on intestinal stem cells [41]. They showed that the regenerative response to ionizing radiation is impaired in the aged murine intestine and that this impairment was caused by reduced regenerative capacity of intestinal stem cells due to diminished Wnt signaling in aged intestinal stem cells, which exhibited reduced expression of stem cell markers. Since the absolute number of intestinal stem cells did not differ between young and aged mice, impaired regenerative capacity in aged mice was a result not of the decreased number of intestinal stem cells, but of reduced stem cell function of aged intestinal stem cells due to reduced Wnt signaling. One of the causes of diminished Wnt signaling in aged intestinal stem cells was reduced Wnt secretion from aged niche cells. The authors also showed that restoration of Wnt signaling by supplementation of Wnt3a to the culture restored the regenerative capacity of aged murine intestinal stem cells, as assessed by organoid formation. Taken together, this study of the aged murine intestine suggested that aging-related dysfunction in the intestine was caused by decreased Wnt signaling in intestinal stem cells.

Another recent study from Pentinmikko and colleagues on Wnt signaling and intestinal stem cells also reported that Wnt signaling was reduced in aged intestinal stem cells [42]. The authors showed that Wnt signaling in intestinal stem cells was diminished by production of Notum, a secreted Wnt inhibitor, by aged Paneth cells. Mechanistically, this secretion of Notum was initiated by activation of mammalian target of rapamycin complex 1 (mTORC1), a protein complex that regulates various intracellular signals [43] and has been reported to be involved in aging [44]. Then, the activated mTORC1 inhibited peroxisome proliferator activated receptor α, which led to increased Notum expression in aged Paneth cells. Notum, secreted from aged Paneth cells, induced inhibition of Wnt signaling in Lgr5+ intestinal stem cells and led to decreased stem cell function, as measured by decreased organoid formation. Genetic deletion of *Notum*, as well as Wnt3a supplementation, restored the function of aged intestinal organoids, and pharmacological inhibition of Notum also restored the regenerative capacity of the aged murine intestine. Hence, this study also indicated that decreased Wnt signaling in intestinal stem cells was the underlying factor in the aging of the intestine.

On the other hand, Cui and colleagues reported the impact of augmented Wnt signaling on aged intestinal stem cells [45]. They showed that intestinal stem cells of 24-month-old mice were incapable of forming typical intestinal organoids with villus and crypt structures, such as is seen in young intestinal organoids. Instead, these cells formed big round sphere-shaped organoids devoid of differentiated cell types. However, the aged intestinal stem cells were able to form typical intestinal organoids when they were cultured in the medium with reduced R-spondin1. The authors concluded that over-responsiveness to Wnt signaling of aged intestinal stem cells mediated the aging-induced deficiency in differentiation. Their report is different from the above two reports in that these authors focused on differentiation rather than regenerative capacity, which may explain the discrepancy between the studies.

The studies above, although through a different mechanism, proposed that Wnt signaling in intestinal stem cells plays an important role in the aging of the intestinal tract. In particular, decreased Wnt signaling in aged intestinal stem cells led to impaired regenerative capacity. However, there are other reports which suggest correlation of enhanced Wnt signaling and aging of the gastrointestinal tract.

One of these studies was performed by Hayashi and colleagues, who investigated the effect of aging on the motility of the stomach. They found that the number of interstitial cells of Cajal (ICC) was decreased in the stomachs of aged mice and that this decrease was associated with impaired gastric compliance [46]. It was found that Wnt signaling was overactivated in the gastric muscles of aged mice and in the gastric corpus muscles of middle-aged humans, and that this overactivation of Wnt signaling inhibited the proliferation of ICC stem cells. Mechanistically, it was shown that TRP53 was upregulated in ICC stem cells when Wnt/β-catenin signaling was constitutively activated, which then downregulated the self-renewal genes, resulting in ICC stem cell growth arrest. Taken together, the results of this study of the aged murine stomach suggested that aging of the gastrointestinal tract was related to enhanced Wnt signaling in the stomach.

Recently, we reported further evidence of a correlation between enhanced Wnt signaling and aging in the stomach [47]. We found that gastric organoids generated from aged mice exhibited increased organoid formation and cellular proliferation compared to gastric organoids generated from young mice. These aged gastric organoids exhibited enhanced Wnt signaling. This enhancement led to the induction of T-box3 (Tbx3), a transcription factor that suppresses cellular senescence [48], in these organoids. We found that cellular senescence in aged gastric organoids was suppressed as a result of enhanced G2-M transition through Tbx3-induced suppression of p19^ARF^, p53, and p21^WAF1^ (Figure 2). Further analysis revealed that the enhancement in Wnt signaling was due to decreased Dkk3, a Wnt inhibitor which has been reported to be correlated with aging-related skeletal muscle atrophy [49]. Investigation of the rationale behind the suppressed Dkk3 revealed that the suppression of Dkk3 in aged gastric organoids resulted from the methylation of their *Dkk3* gene. Hence, consistent with the report from Hayashi and colleagues, we observed correlation of enhanced Wnt signaling and aging of the gastrointestinal tract.

Taken together, the studies described above indicate that Wnt signaling plays an important role in the aging of the intestine and the stomach.

## 5. Wnt Signaling and Aging-Related Carcinogenesis of the Gastrointestinal Tract

Aging is considered one of the risk factors for cancers in various organs [1]. Regarding the role of Wnt signaling in aging-related carcinogenesis, Tao and colleagues reported that enhanced Wnt signaling increased sensitivity to DNA damage in intestinal stem cells located in the base of the intestinal crypts, which can be involved in the initiation of cancers [50]. As we age, DNA mutations accumulate in stem cells and the clonal expansion of daughter cells that possess these mutations can potentially lead to cancer initiation and progression [51].

Epigenetic reprogramming in stem cells has also been reported to induce malignant neoplasm in aging stem cells [52], and methylation of tumor suppressor genes have been reported to increase with age in normal gastric mucosa [53]. Similarly, methylation of *TCF4*, a transcription factor that functions as the downstream effector of canonical Wnt/β-catenin signaling in normal gastric mucosa was reported to increase with age [54]. The report from Kim and colleagues also showed that knocking down *TCF4* enhanced proliferation of cells in in vitro studies using a gastric cancer cell line, showing that Wnt signaling altered proliferation in aged gastric mucosa.

In our recent study, we showed that DNA methylation of the *Dkk3* gene in aged gastric organoids, the cells originating from the aged gastric mucosa, led to enhanced Wnt signaling and Tbx3 induction, resulting in increased proliferation due to suppressed cellular senescence [47]. Regarding the Wnt-dependent enhanced proliferation observed in aged gastric organoids, Milavonic and colleagues reported a suggestive phenomenon [55]. They found that B-cell lymphoma cells that were once senescent and were released from senescence re-entered the cell cycle and showed Wnt-dependent enhanced clonogenic proliferation. Interestingly, this clonogenic proliferation was not observed in cells that were never senescent. It is possible that the same phenomenon is involved in clonogenic expansion of Dkk3-suppressed aged gastric organoids. In addition to the augmented cellular proliferation seen in the aged murine gastric organoids, analysis of human gastric tissues revealed that the expression of TBX3 in the gastric mucosa increased with the age of patients, whereas that of DKK3 showed a reciprocal trend. Similarly, TBX3 expression was increased in precancerous changes of the gastric mucosa and increased even more in gastric cancer tissues. These findings suggest a possible Wnt-related molecular mechanism underlying the aging-related carcinogenesis of the stomach.

## 6. Future Perspectives

The studies discussed above strongly indicate that the Wnt signaling pathway is involved in the aging of the gastrointestinal tract. In addition, our study has suggested a possible mechanism contributing to aging-related carcinogenesis in the stomach, namely, the Dkk3-Wnt-Tbx3 pathway [47]. Since this pathway was involved in the carcinogenesis of human gastric cancers, it can be considered a candidate for a therapeutic target in aging-related gastric cancers. The genetic characteristic underlying the aged gastric organoids was suppression of Dkk3, and, since supplementation of recombinant Dkk3 was able to repress the augmented cellular proliferation in aged gastric organoids, DKK3 could be considered a potential therapeutic reagent for gastric cancers of the elderly. Another potential therapeutic target of the Dkk3-Wnt-Tbx3 pathway is the Wnt target gene Tbx3, whose overexpression has been reported to be correlated with the advanced tumor stage of human gastric cancers [56].

Transcription factor TBX3 was first identified as the responsible gene for ulnar-mammary syndrome, a rare pleiotropic disorder affecting limb, apocrine gland, tooth, and genital development [57]. It functions as a suppressor of cellular senescence by regulating cell cycle related genes, as mentioned earlier [48]. Overexpression of TBX3 has been reported in various cancers, such as melanomas, where TBX3 has been reported to repress E-cadherin and enhance their invasiveness [58]. The overexpression of TBX3 has also been reported in breast cancers [59] and TBX3 was reported to promote progression of non-invasive breast cancers to invasive cancers [60]. Due to the pivotal role it plays in the progression of breast cancers, TBX3 has been considered as one of the therapeutic drug targets in breast cancers [61], and the regulation of TBX3 in breast cancers has been intensively studied. Amir and colleagues reported that microRNA-206 suppressed TBX3 and, as a result, inhibited tumor growth of breast cancer organoids [62], which suggests the possibility that this microRNA could also regulate TBX3 in aging-related gastric cancers. With respect to the chemical inhibitor of TBX3, Paul and colleagues recently discovered that two alkaloids Jervine and Diflomotecan were able to form stable complexes with TBX3 through an integrated computational approach and that these two alkaloids could become new effective drugs for breast cancers [63]. Given their effect on TBX3, these two alkaloids may also be effective in treating aging-related gastric cancers.

Other Wnt signaling inhibitors have also been considered as therapeutic reagents for solid tumors, including gastric cancers and colorectal cancers, and hematological malignancies [64,65]. WNT974 which inhibits Porcupine, the molecule required for secretion of Wnt, is now in phase 2 of a clinical trial for head and neck squamous cell cancers. The β-catenin antagonist PRI-724, together with leucovorin calcium, oxaliplatin, or fluorouracil, are also in phase 2 of a clinical trial for colorectal cancers. These reagents could be strong candidates for future chemotherapy against these cancers. However, Wnt signaling is involved in myriad aspects of homeostasis and inhibiting gross Wnt signaling could cause various unexpected adverse effects. Hence, not targeting pan-Wnt signaling but, instead, specific downstream molecules, such as TBX3, may represent a more promising therapeutic approach for aging-related gastric cancers in clinical practice.

## 7. Conclusions

As discussed in this review, it is evident that Wnt signaling plays a crucial role in the aging of the gastrointestinal tract. However, whether enhanced or diminished Wnt signaling is related to aging is still controversial. It is possible that it depends on the organs involved, such as observed in Wnt-suppressed aged neural stem cells and Wnt-enhanced aged muscular stem cells, but further studies are awaited to elucidate this issue.

## Figures and Tables

**Figure 1 ijms-23-12210-f001:**
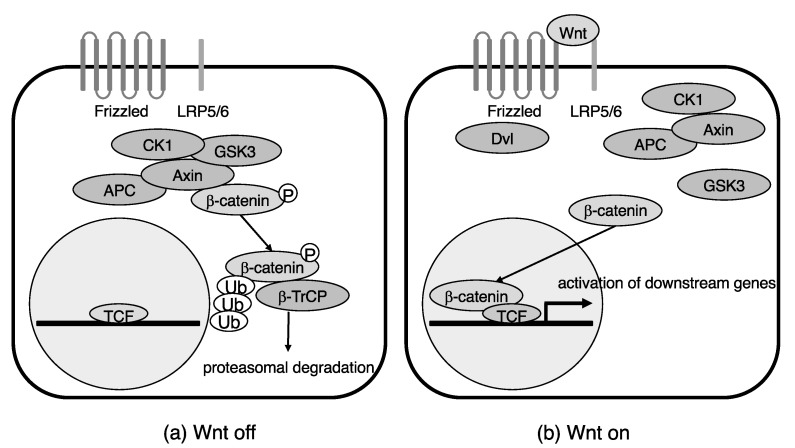
Canonical Wnt signaling pathway. (**a**) Cytosolic β-catenin is subjected to proteasomal degradation in the absence of Wnt signaling. (**b**) Upon Wnt binding to its receptor, β-catenin translocates into the nucleus and activates its downstream genes. Ub: ubiquitin.

**Figure 2 ijms-23-12210-f002:**
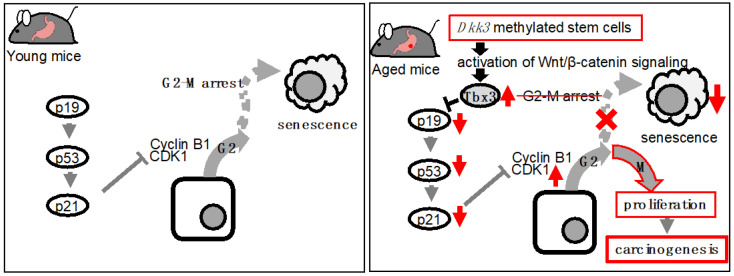
Induction and evasion of cellular senescence in young and aged gastric stem cells. (**a**) Gastric stem cells of young mice undergo cellular senescence via G2-M arrest. (**b**) *Dkk3* methylated aged stem cells evade cellular senescence via G2-M transition as a result of Tbx3 induction.

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
