# Peer review of "Wnt Signaling and Aging of the Gastrointestinal Tract"

_ijms, 2022, doi:10.3390/ijms232012210_

Round 1
Reviewer 1 Report
In the submitted manuscript "Wnt Signaling and Aging of the Gastrointestinal Tract", the authors briefly but in sufficient detail described the Wnt signaling pathway and its involvement in the aging process of individual cells, including cells of the gastrointestinal tract. Separately, the authors focused on the role of the Wnt signaling pathway in carcinogenesis and possible dysregulation of the Wnt signaling pathway leading to the transition of a stem cell from normal aging to tumorigenesis.
The manuscript needs some minor editing in English. For example, line 162 and line 184, instead of Present Continuous "is playing" it is better to use Present Simple "plays".
Author Response
The authors would like to thank the reviewer for reviewing our manuscript and for his/her comments.
We have carefully checked our manuscript and made corrections where needed.
We really appreciate the reviewer for taking his/her time and helping us improve our manuscript.

Reviewer 2 Report
Asano et al highlighted the effect of Wnt signaling in the aging of gastrointestinal tract and aging-related carcinogenesis. Overall, the review is well performed, but there are little minor points the author would need to address.
Minor points:
1. As Wnt signaling plays a controversial role in regulating aging. Discuss some potential therapy targets (inhibitor or agonist) or hypothesis for the future research on Wnt signaling regulating the aging of gastrointestinal tract and aging-related carcinogenesis.
Author Response
We thank the reviewer for reviewing our manuscript and for his/her comments.
As proposed by the reviewer, we have added a new section "Future Perspectives" which discusses about potential Wnt signaling-related therapy for aging-related gastric cancers.
We really appreciate the reviewer for taking his/her time and helping us improve our manuscript.
